# Assessing Discharge Readiness After Propofol-Mediated Deep Sedation in Pediatric Dental Procedures: Revisiting Discharge Practices with the Modified Aldrete Recovery Score

**DOI:** 10.3390/children12091155

**Published:** 2025-08-29

**Authors:** Merve Hayriye Kocaoglu, Cagil Vural

**Affiliations:** 1Department of Pediatric Dentistry, Faculty of Dentistry, Ankara University, 06560 Ankara, Türkiye; 2Department of Oral and Maxillofacial Surgery, Faculty of Dentistry, Ankara University, 06560 Ankara, Türkiye

**Keywords:** pediatric sedation, procedural sedation, discharge, modified aldrete recovery score, non-operating room anesthesia, recovery assessment, propofol

## Abstract

**Background**: Efficient and safe discharge is critical in pediatric dental procedures performed under deep sedation in non-operating room anesthesia (NORA) settings. Traditional institutional criteria may delay discharge due to subjectivity. Objective: This study compared the Modified Aldrete Recovery Score (MAS) and institutional discharge criteria to determine which provides faster and reliable discharge decisions. **Methods**: In this prospective observational study, 100 children (ages 2–10, ASA I–III) undergoing deep sedation for dental treatment were evaluated. Two nurse anesthetists independently assessed discharge readiness every five minutes using either MAS or institutional criteria. Demographic data, BMI percentile, ASA class, anesthesia duration, and propofol dose were recorded. Discharge times were compared using Wilcoxon signed-rank and subgroup analyses and correlation tests. **Results**: MAS allowed significantly earlier discharge than institutional criteria (24.75 ± 7.33 vs. 36.79 ± 8.59 min, *p* = 0.01). The agreement between methods was poor (ICC = 0.06). Discharge time varied significantly by BMI percentile (*p* = 0.01); obese children had shorter recovery times, while time differences were greater in overweight children. No adverse events or readmissions occurred. **Conclusions**: MAS provides a quicker and equally safe discharge assessment in pediatric dental sedation. Its use may enhance workflow efficiency and standardize recovery decisions in NORA settings lacking formal PACUs.

## 1. Introduction

Behavioral modification is the first-line approach to managing fear and pain in pediatric dentistry, combining communication and cognitive-behavioral strategies with environmental adaptation to improve cooperation and reduce distress. Core methods—tell-show-do, modeling, positive reinforcement, voice control, and parental presence/absence—are complemented by distraction modalities (music, games, video/TV, VR) and preparatory counseling; recent evidence suggests music, aromatherapy, and game-based distraction rank among the most effective non-pharmacological options for reducing dental anxiety in children. In randomized comparisons, techniques such as tell-show-do, modeling, and yogic relaxation have each demonstrated anxiety reduction during pediatric dental care, underscoring the value of structured behavioral programs [1,2]. Effective pain control (topical and local anesthesia delivered with child-centred communication) is integral to these strategies; when non-pharmacological measures are insufficient, nitrous oxide–oxygen minimal sedation is a safe and effective adjunct for anxious yet otherwise healthy children. For patients with severe anxiety, extensive treatment needs, special health care needs, or failed office-based behavior guidance, escalation to deeper sedation or general anesthesia may be indicated, provided that monitoring, staffing, and discharge criteria meet established pediatric safety guidelines [3].

Due to the increased demand for dental restorations for children under sedation, various types of restorations are often carried out in operating rooms, non-operating room anesthesia (NORAs) or dental practices [4]. In cases where direct discharge to home is mandatory, the risk of residual sedation should be well managed [5,6]. However, space limitations, a shortage of personnel, and the absence of structured post-anesthesia care units (PACUs) can restrict recovery monitoring in such environments. In this context, implementing a safe and efficient discharge protocol is crucial.

Many regulatory organizations and accreditation committees worldwide have put forward policies and procedures for the safe recovery and discharge of patients after anesthesia [7,8]. The American Society of Anesthesiologists (ASA) and American Association of Pediatric Dentists (AAPD) sedation guidelines stipulate that the patient should have returned to the level of alertness before the procedure at the time of discharge [3,9]. The most common adverse event that may develop after sedation is residual sedation, which usually develops after leaving monitored care [3]. Some centers still use discharge decisions based on institutional criteria or clinical judgment. However, such decision-making regarding discharge may lead to differences in practice, may not capture the presence of residual sedation, and may lead to unnecessary waiting time to eliminate this risk. This time-dependent discharge approach, which is based on the patient returning to their pre-anesthesia physical and cognitive capacity, can result in very long waiting times, which can disrupt the effective functioning of the sedation unit and its team [10]. Alternatively, scales such as the Aldrete Post-Anesthesia Recovery Scale, introduced in 1970, which use physiological parameters to test recovery after anesthesia, provide more objective discharge decisions by eliminating subjective judgement [11,12,13]. With the concept of ambulatory anesthesia and the widespread use of procedural sedation, a modification was needed in the recovery scale as ‘*home readiness or street fitness*’ of the patient [14]. In the Modified Aldrete Recovery Score, recovery corresponds to a score of 9/10. Each parameter is scored between 0 and 2, with a maximum score of 1 in only two of them. It has also been revised for ambulatory surgery and can be adapted to the type of procedure performed. Adequate scores in physiological parameters in this scale may indicate that the patient can be discharged to the ward or directly home after outpatient anesthesia procedures [14].

Different institutional or traditional approaches are applied depending on the characteristics of the patient and time when assessing readiness for discharge after sedation [15,16,17,18]. In many centers where pediatric anesthesia is performed, various scales are used to confirm recovery, such as transition from Phase 1 recovery unit (PACU) to Phase 2 recovery for general anesthesia or direct discharge home. However, there are limited studies in the literature comparing recovery and discharge readiness after pediatric, dental deep sedation in terms of clinical assessment and recovery scores. Therefore, we hypothesized that the Modified Aldrete Recovery Score enables faster discharge after deep dental pediatric sedation than institutional discharge criteria and is sufficiently safe. The primary aim of this study was to compare the discharge times according to the two methods. The secondary aim was to investigate the relationship between discharge times measured by both methods and patient demographic and physical characteristics, duration of anesthesia and total amount of propofol used.

## 2. Materials and Methods

### 2.1. Study Design and Patient Population

Following approval from the institutional ethics committee (approval number: 36290600/73/2022) and the acquisition of informed consent from all patients or their relatives, a total of 104 patients were included in this prospective, observational study between January 2024 and June 2024. All procedures applied in this study were carried out in accordance with the Declaration of Helsinki. In our institution, all procedures performed under deep sedation are conducted in the sedation unit within the Ankara University Department of Pediatric Dentistry. The inclusion criteria were children between 2 and 10 years of age with an American Society of Anesthesiology (ASA) (Appendix A) physical status I–III, “definitely negative and negative” (rating 1 and 2, respectively) according to Frankl Behavior Rating Scale, who could not be treated with any behavior modification method and therefore who had sedation indication. Exclusion criteria were being younger than 2 years of age or older than 10 years of age and ASA IV; having an history of allergy to anesthetic drugs or having muscle, kidney or liver disease that may alter anesthetic drug metabolism.

### 2.2. Power Analysis

As there were no similar studies in the literature, the sample size was calculated based on the effect size. Taking the effect size of the time difference between the MAS and institutional discharge criteria as 0.4 and using the paired-*t* test with a power of 0.80 at a significance level of 0.05, the minimum required sample size was found to be 84. However, to allow for potential losses during the study, it was planned to include 104 patients.

### 2.3. Statistical Analyzes

Descriptive statistics for continuous variables were presented as mean ± standard deviation (X ± SD). The normality of data distribution was assessed using the Kolmogorov–Smirnov test. Since the data were not normally distributed (*p* = 0.01) and some groups had small sample sizes, non-parametric statistical methods were used. The Mann–Whitney U test was applied for group comparisons based on gender and comorbidity, and the Kruskal–Wallis test was used for comparisons across BMI and ASA categories. The Wilcoxon signed-rank test was performed to compare paired discharge time measurements (MAS vs. institutional discharge criteria). Spearman’s rank correlation analysis was conducted to evaluate associations between recovery times and continuous variables such as age, weight, height, BMI, BMI percentile, and total propofol dose. Statistical significance was defined as *p* < 0.05. All analyses were conducted using SPSS version 25.0.

### 2.4. Anesthesia Procedures

All patients were monitored by pulse oximetry, electrocardiography, non-invasive blood pressure (NIBP), (Dräger, Infinity Vista XL monitor, Lübeck, Germany), capnography (Capnostream 35, Medtronic, Inc., Dublin, Ireland). In order to establish a painless intravenous line for the pediatric patient, anesthesia induction was performed for a few minutes using mask ventilation with a 1–8% sevoflurane solution in an oxygen-air mixture (2 L/min). Once the patient became unresponsive, intravenous access was established in the back of the hand. After this stage, inhalation anesthesia was discontinued. Intravenously, 0.05 mg/kg midazolam, 0.5 mg/kg lidocaine and 1 mg/kg propofol were administered. According to the ASA’s “Statement on the continuum of depth of sedation, the definition of general anaesthesia, and the levels of sedation and analgesia”, deep sedation anaesthesia was maintained with a propofol infusion (Braun Perfusor Space™; B. Braun, Melsungen, Germany) within a BIS range of 50–60. During the procedure, continuous oxygen flow was provided at a rate of 2–3 L/min through a nasal cannula in semi-sitting position. Paracetamol 10 mg/kg was administered to all patients as standard analgesia protocol. In accordance with ASA and APA/AAPD guidelines, 2 anesthetists with training and skills in pediatric advanced life support were present throughout the sedation [3]. In addition to the primary anesthetist, who performed all interventional procedures, a nurse anesthetist was responsible for monitoring the vital signs throughout sedation. Restorative procedures (fissure sealants, glass ionomer restorations, compomer and composite resin restorations, pulpotomies and pulpectomies, stainless steel crowns and strip crowns) were performed by the pediatric dentist in all patients.

### 2.5. Evaluation of Discharge Scores

The patient was awakened in the sedation unit after the procedure and transferred to the recovery room; at this point, the timer was started. Patients were continuously monitored for oxygen saturation and blood pressure via NIBP cuff. One nurse anesthetist assessed the same patient for recovery using institutional discharge criteria, while another nurse anesthetist, who had previously been trained to perform MAS, assessed the patient. The two nurse anesthetists, who were unaware of each other’s assessments, independently evaluated the patient’s recovery every five minutes using two different methods. When the patient had a score of 9/10 according to the MAS, the nurse anesthetist determined that they were “recovered” and “ready for discharge”, and confidential notes were made. Meanwhile, the other nurse anesthetist assessed the monitored patient and recorded ‘ready for discharge’ as soon as they fully met each item of the institutional discharge criteria (Figure 1). To prevent bias between the two administrators, a third observer (an anesthetist) checked the notes of both evaluators. After checking the evaluators’ notes, the external observer terminated the study according to whichever method resulted in a more delayed discharge decision. Demographic data including age, gender, BMI, BMI-Percentile, ASA, and presence of comorbidities were recorded. In this study, the use of BMI in children was categorized using BMI percentiles, as BMI alone is insufficient for evaluating children’s growth and development. In this calculation, the CDC BMI percentile calculator (https://www.cdc.gov/bmi/child-teen-calculator/index.html, accessed on 1 May 2025) was used to categorize patients into underweight, normal, overweight, and obese categories based on their calendar age, gender, height, and weight. Total duration of anesthesia (min), total propofol dose (mg), recovery time according to MAS (minutes) and recovery time according to institutional discharge criteria (minutes) were recorded. Recovery time was compared in all patients in terms of both methods. The effect of demographic parameters on patient discharge times was also analyzed (Figure 2).

Patients with scores of 9–10 can be discharged; a score of 8 or lower indicates the need for continued close observation.

## 3. Results

Four patients were not included: one because of bleeding from extraction socket that required extended post-anesthesia monitoring, and three because persistent desaturation prevented the surgery from being completed. A total of 100 children (55% male, 45% female) were included in the study. The majority had a normal BMI (63%), while 23% were underweight, 11% overweight, and 3% obese. Most patients were classified as ASA I (71%), and 29% had comorbidities. 7 patients were classified as ASA III and Those were with following comorbidities.

Diabetes mellitus + AsthmaDown Syndrome + HypothyroidismAortic stenosisIntellectual Disability + Epilepsia + Hypoxic Brain InjuryAtrial Septal Defect + Thorax deformationAtrial Septal Defect + Aortic RegurgitaitonSevere F7 Deficiency

The mean age was 5.52 ± 1.75 years, with a mean BMI of 15.2 ± 1.95 kg/m^2^ and a BMI percentile of 38.19 ± 33.76 (Table 1). The average duration of anesthesia was 63.99 ± 16.52 min, and the mean propofol dose was 298.23 ± 112.64 mg (Table 2).

The mean discharge time for patients according to the MAS was 24.75 ± 7.33 min, whereas the discharge decision was made in 36.79 ± 8.59 min according to the institutional discharge criteria (Table 3). MAS time and institutional discharge time were statistically different from each other (*p* = 0.01, *p* < 0.05). In addition, an intraclass correlation analysis was conducted to examine the consistency in the relationship of both durations, and it was found that the two durations were not significantly consistently correlated (rICC = 0.06, *p* = 0.36) (Table 3). In summary, the durations are independent and different from each other. In this context, since the discharge according to the institutional criteria was higher, the difference in (discharge according to the institutional criteria—(MAS time) was taken and included in the analyses.

In the study, it was found that there was no statistically significant difference when MAS time and inter-method time difference were compared in terms of gender (*p* > 0.05). However, institutional discharge criteria duration differed according to gender and was found to be higher in male patients (*p* = 0.02, *p* < 0.05). MAS duration, institutional discharge criteria duration and duration difference between two methods were found to be significantly different according to BMI percentile groups (*p* = 0.01). While the MAS and institutional discharge criteria measurements of obese individuals were found to be shorter than those of underweight, normal and overweight individuals, the time difference between the two methods was significantly higher in overweight individuals compared to other groups (Table 4).

Age, height, weight, BMI, BMI-percentile, duration of anesthesia, propofol dose values of the patients were not associated with MAS time, institutional discharge time and intermethod time difference (*p* > 0.05) (Table 5).

One patient with a normal BMI percentile experienced desaturation (SpO_2_ 88%) and recovery delirium, resulting in a discharge time of 50 min. An overweight patient developed bronchospasm (SpO_2_ 84%), which was resolved with mask ventilation. Another patient with tonsillar hypertrophy experienced transient desaturation (SpO_2_ 90%), which was also managed with no further complications. All patients (*n* = 100) were safely and successfully discharged home the same day, with no serious emergencies or unexpected readmissions reported.

## 4. Discussion

In this prospective, observational study of pediatric patients undergoing deep dental sedation in a non-operating room setting, it was found that the discharge decision based on institutional discharge criteria took longer than the discharge decision based on the MAS. The findings revealed a statistically significant difference in discharge times between the two methods. MAS permitted notably earlier discharge decisions than the institution’s standard clinical approach. This lends weight to the notion that standardized scoring systems could facilitate more efficient patient flow without compromising safety.

As there is no significant intraclass association between MAS and institutional discharge durations, these two approaches likely function independently and may not always coincide in determining discharge readiness. This results in an irrelevant PACU length of stay (LOS). In practical terms, this highlights potential overcautiousness or variability in traditional clinical judgement compared to objective, criteria-based evaluations, such as the MAS. Considering the increasing importance of efficiency and safety in non-operating room anesthesia (NORA) settings, this finding underscores the importance of structured discharge tools. In procedural sedation and outpatient day surgery facilities rapid patient turnover is just as important as ensuring safe discharge [19,20]. Standardized discharge practices—such as using single short-acting agents and validated scoring systems—are essential for ensuring both efficiency and patient safety in outpatient procedures [17,21,22]. Replacing subjective clinical judgment with objective scales enhances precision in assessing recovery and discharge readiness [19]. The development of any scale involves a few steps aimed at ensuring the validity and reliability of the scale [23]. The Aldrete Score, although independent of drug type or sedation depth, has been validated for use after general anesthesia and outpatient surgery, and can be modified for different clinical settings [14]. It was also emphasized that this score could be modified by considering different parameters in different surgical and anesthetic settings [14]. APA/AAPD guidelines recommend using standardized, validated recovery scores—applied without interpretation by trained staff—to confirm a child’s return to pre-sedation physical and cognitive function [3].

In our study, both methods that confirmed recovery in the same subject were tested and the clinical assessment time for discharge decision was found to be statistically significantly longer. In addition, there was no internal consistency in recovery times between methods. Here, the decision to discharge patient home can be influenced by the physiological parameters reached, as well as by the length of time elapsed. The presence of a nurse in the recovery area or regular checks of the patient under the supervision of a caregiver can also lead to different interpretations. In this study, the recovery area was continuously monitored by two nurse anesthetists, and it was aimed to avoid any delay in the observation of recovery. However, controlling the patients in the recovery area at certain time intervals such as every 10 min may involuntarily prolong the discharge time from the recovery area [24].

On the other hand, contrary to the inconsistency between the two methods, the recovery times and the difference in time between the two methods are similar in terms of age, gender and ASA groups. A study conducted by Truong et al. in 2004 with 292 subjects showed that the use of modified Aldrete score shortened the duration of PACU LOS [15]. The fact that the recovery time was similar between age and ASA groups for both methods in our study may be attributed to careful patient selection before sedation as recommended by the guidelines. In the literature, comorbidities have been shown to increase sedation-related complications and in addition, more debilitated patients are associated with unexpectedly longer recovery time [25,26]. Another noteworthy aspect of our study is the homogeneity of the recovery time between patients, which once again emphasizes the consistency of the MAS in appropriate patient selection and appropriate sedation indications.

In this study, pediatric patients were stratified by BMI percentile (underweight, normal weight, overweight, obese) to examine how body habits relate to recovery and discharge. When discharge metrics were compared across BMI groups, obese children had the shortest times by both measures: MAS (Modified Aldrete Score) recovery time 18.33 ± 7.64 min and institutional discharge time 28.33 ± 10.41 min (*p* = 0.01 for the between-group comparison). The interval between “MAS ready” and institutional discharge—the MAS–institutional discharge difference—was greatest in the overweight group (15.45 ± 7.89 min; *p*: 0.01), indicating a larger delay from objective readiness to final discharge in this subgroup.

Although obesity is commonly expected to prolong emergence from anesthesia due to OSA-related airway obstruction and potential re-sedation from adipose tissue redistribution [24], our cohort showed the opposite pattern: both MAS-based recovery and institutional discharge were shorter in obese/overweight children. This observation is consistent with a large pediatric sedation cohort (*n* = 28,792), in which delayed recovery occurred 2.66 times more frequently in non-obese patients [25]. A plausible pharmacokinetic explanation is that children with less adipose tissue and a smaller peripheral compartment (i.e., underweight/normal BMI) may sustain higher circulating propofol concentrations for longer, leading to prolonged MAS recovery despite the theoretical risk of delayed recovery in obesity due to redistribution [27,28].

In our study, the mean discharge time was 36.79 ± 8.59 min by institutional discharge criteria and 24.75 ± 7.33 min by MAS, respectively (*p* = 0.01). In a study conducted by Malviya et al. in sedated pediatric patients for echocardiography, the time to complete recovery according to the standard clinical practice of the recovery nurse was found to be earlier according to the Modified Ability to Maintain Alertness Test. However, when the depth of sedation was controlled by Bispectral Index (BIS) monitoring, it was observed that only 55% returned to the initial baseline BIS value when the nurse decided that the patient could be discharged. Baseline BIS values were achieved in 92% of the patients who were recovered according to the recovery scale (*p* < 0.05). The conclusion that can be drawn from that study is that the application of a combined awakening scale better guarantees a return to baseline BIS but prolongs the traditional waiting time [22]. In addition, according to the results of a study in the literature investigating the effectiveness of BIS use during dental deep sedation, it was shown that the use of BIS did not change the wake-up time but significantly shortened the discharge time from the PACU [29]. Our study stresses that a faster and more uneventful discharge can be achieved with the MAS after dental pediatric deep sedation, preferably combined with a BIS monitor, compared to clinical observation.

The aim of our study was to show that patients can be ready for discharge home after deep sedation mediated with propofol as soon as possible with the MAS in non-operating room procedures such as dental procedures. Most NORA settings do not have PACU facilities. Therefore, it is crucial to capture the state of the pediatric patient when they are most fit to go home when they come out of monitored care. In the literature, there are studies showing that it is safe to shorten the PACU stay or skip it directly after day surgery, provided that the recovery is ensured by scales [21,30]. In a study evaluating recovery after gastroscopy in adult subjects, recovery times were compared after non-anesthetist caregivers were given MAS training. According to the results of that study, it was found that the departure time from the recovery area was shortened by 19–35% compared to their institutional method [31]. Our study makes a valuable contribution to the existing literature by demonstrating the safe and efficient discharge of children from the recovery area following propofol-mediated deep sedation for dental procedures in NORA settings. Other agents and their combinations with propofol may be the subject of future studies in terms of safety, time, and the effectiveness of sedation unit operations.

### Limitations

A key strength of this study is its prospective design and the inclusion of a well-defined pediatric outpatient cohort. However, several limitations should be acknowledged. First, although the sample included ASA I–III patients, no complications occurred during or after sedation. As such, the study may be underpowered to assess the safety of deep dental sedation and post-discharge outcomes in higher-risk groups, such as ASA IV patients with poor physical status. Second, BIS monitoring was discontinued after the completion of dental procedures, and no correlation was made between BIS values and recovery level. This was primarily due to two factors: pediatric patients often become agitated upon awakening and may remove monitors, including the BIS sensor; and continued display of BIS values could have biased the observer’s judgment during subjective recovery assessment. Although the monitor could have been blinded, this remains a methodological limitation. Third, as a single-center study, the findings reflect one institution’s discharge protocol. Practices vary widely across centers; in some settings, patients may be discharged directly from the operating room without a dedicated recovery area. These differences were not captured in our analysis and may limit the generalizability of our findings. Lastly, this study did not include a cost-effectiveness analysis comparing the two discharge methods. Evaluating the economic impact of earlier discharge based on objective criteria remains an important area for future research.

## 5. Conclusions

The modified Aldrete recovery score provides a faster and more consistent alternative to institutional clinical criteria for assessing the discharge readiness of children following propofol-mediated deep sedation. Its implementation in pediatric NORA settings may reduce discharge time variability and improve workflow efficiency without compromising safety.

## Figures and Tables

**Figure 1 children-12-01155-f001:**
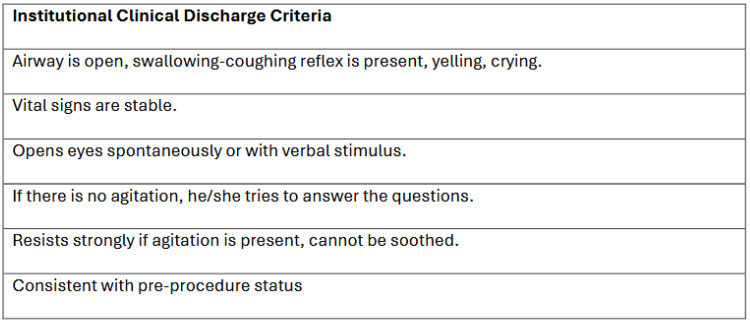
Institutional Clinical Discharge Criteria.

**Figure 2 children-12-01155-f002:**
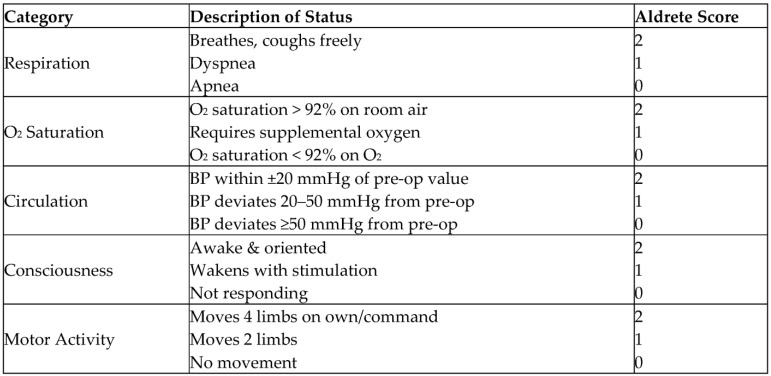
Modified Aldrete Recovery Score.

**Table 1 children-12-01155-t001:** Demographic data.

Demographic		*n* (%)
Gender	Male	55 (55)
	Female	45 (45)
BMI-Percentile	Normal	63 (63)
	Obese	3 (3)
	Overweight	11 (11)
	Underweight	23 (23)
Comorbidity	Yes	29 (29)
	No	71 (71)
ASA	I	71 (71)
	II	22 (22)
	III	7 (7)

**Table 2 children-12-01155-t002:** Patient measurements.

Variable	X ± S.D.
Age (yr)	5.52 ± 1.75
Anesthesia Duration (min)	63.99 ± 16.52
Propofol (mg)	298.23 ± 112.64

yr: years, min: minutes, mg: miligrams.

**Table 3 children-12-01155-t003:** Recovery Times for Different Methods.

Measurements	X ± S.D.	*p*
MAS Time (min)	24.75 ± 7.33	0.01 *
Institutional Discharge Criteria Time (min)	36.79 ± 8.59	

* Wilcoxon signed-rank test—Intraclass correlation (significant difference at 0.05 level).

**Table 4 children-12-01155-t004:** Times According to Gender, BMI-Percentile Group, Comorbidity and ASA Scores.

		MAS Time	*p*	Institutional Discharge Criteria Time	*p*	Inter-Method Time Difference	*p*
		(X ± S.D.)		(X ± S.D.)		(X ± S.D.)	
Gender	Male	25.87 ± 7.19	0.14	38.71 ± 8.94	0.02 *	12.84 ± 7.79	0.22
	Female	23.38 ± 7.36		34.44 ± 7.61		11.07 ± 7.24	
BMI-Percentile Groups	Normal	25.59 ± 6.91	0.01 *	37.14 ± 9.12	0.01 *	11.56 ± 7.29	0.01 *
	Obese	18.33 ± 7.64		28.33 ± 10.41		10.00 ± 5.01	
	Overweight	22.27 ± 8.47		37.73 ± 6.07		15.45 ± 7.89	
	Underweight	24.48 ± 7.66		36.48 ± 7.79		12.00 ± 8.37	
Comorbidity	Yes	24.52 ± 8.09	0.72	38.26 ± 10.32	0.34	13.74 ± 7.39	0.14
	No	24.86 ± 7.03		36.13 ± 7.69		11.28 ± 7.57	
ASA	I	25.21 ± 7.35	0.62	36.28 ± 7.76	0.75	11.07 ± 7.56	0.13
	II	23.55 ± 7.13		37.68 ± 9.73		14.14 ± 7.21	
	III	23.86 ± 8.41		39.14 ± 13.11		15.29 ± 7.45	

Mann–Whitney U test and Kruskall Wallis test (* Significant difference at 0.05 level).

**Table 5 children-12-01155-t005:** Relationships between Age, Weight, Height, BMI, BMI-Percentile, Anesthesia Duration, Propofol dose and Time Periods.

		MAS Time (min)	Institutional Discharge Criteria Time (min)	Inter-Method Time Difference (min)
Age	r	0.05	0.01	−0.04
	*p*	0.63	0.93	0.71
Weight	r	0.01	−0.04	−0.05
	*p*	0.94	0.71	0.62
Height	r	0.11	0.01	−0.09
	*p*	0.29	0.90	0.38
BMI	r	−0.15	−0.11	0.03
	*p*	0.13	0.29	0.79
BMI Percentile	r	−0.15	−0.09	0.04
	*p*	0.14	0.36	0.70
Anesthesia Duration (min)	r	−0.07	0.05	0.13
	*p*	0.51	0.59	0.21
Propofol (mg)	r	0.01	0.09	0.09
	*p*	0.90	0.37	0.37

Spearman’s rank correlation analysis.

## Data Availability

The data presented in this study are available on request from the corresponding author. Due to legal reasons medical data could not be directly published.

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
