# Peer review of "Assessing Discharge Readiness After Propofol-Mediated Deep Sedation in Pediatric Dental Procedures: Revisiting Discharge Practices with the Modified Aldrete Recovery Score"

_children, 2025, doi:10.3390/children12091155_

Round 1
Reviewer 1 Report
Comments and Suggestions for Authors
Dear Authors,
thank you for the article. Here are some suggestions how to improve it:
- In the introduction, you should write more about managing pain and fear - please write more about the behavioural methods to reduce Child's fears. Also, add the information on general anaesthesia as one of the methods of managing fear. Find interesting articles, eg. doi: 10.17219/dmp/169966
- When describing ASA criteria, please draw a graph or add a table on the criteria (could be in Suppl. files)
- Add the information on congenital defects (if they were present)
- Line 157- why do you even consider BMI in children? This is an "adult only" index, besides it is very disputative. Please, find another way to compare weight and height of children. Maybe use percentile grids instead?
- A lot of discussion bases on BMI-index, which is actually incorrect in children. The Authors should change this parameter.
Thank you
Author Response
- In the introduction, you should write more about managing pain and fear - please write more about the behavioural methods to reduce Child's fears. Also, add the information on general anaesthesia as one of the methods of managing fear. Find interesting articles, eg. doi: 10.17219/dmp/169966
Answer: Thank you for your comment. A paragraph to identify pain management and behavior modification methods are added to the introduction section. (Line 31-50) Also, article mentioned is cited for the sake of discussion. Thank you for valuable contribution.
- When describing ASA criteria, please draw a graph or add a table on the criteria (could be in Suppl. files)
Answer: Thank you for your comment. A simplified and more practical version of ASA classification table is added as Supplement 1.
- Add the information on congenital defects (if they were present)
Answer: Thank you for your contribution. There were 7 patients classified as ASA 3.
Those were with following comorbidities. This is mentioned in the manuscript. (Line 198-206)
- Diabetes mellitus + Asthma
- Down Syndrome + Hypothyroidism
- Aortic stenosis
- Mental Retardation + Epilepsia + Hypoxic Brain Injury
- Atrial Septal Defect + Thorax deformation
- Atrial Septal Defect + Aortic Regurgitaiton
- Severe F7 Deficiency
- Line 157- why do you even consider BMI in children? This is an "adult only" index, besides it is very disputative. Please, find another way to compare weight and height of children. Maybe use percentile grids instead?
Answer: Thank you for your comment.
While weight and height percentiles alone can provide information about a child's overall health or growth, they do not provide sufficient information about fat compartment distribution for drug dose titration in anesthetic practice. In fact, BMI percentiles appropriate for age and gender, which are valid criteria used by the CDC, NHS, and WHO, are the preferred method for children. A previous publication on this topic was published in the MDPI-Children journal by Valerio et al. (Valerio, G.; Di Bonito, P.; Di Sessa, A.; Ballarin, G.; Calcaterra, V.; Corica, D.; Faienza, M.F.; Franco, F.; Licenziati, M.R.; Maffeis, C.; et al. Severe Obesity Defined by Percentiles of WHO and Cardiometabolic Risk in Youth with Obesity. Children 2024, 11, 1345. )https://doi.org/10.3390/children11111345
While height and weight percentiles can be added to the study if desired, we believe that the BMI percentile values currently included in our methodology are more appropriate.
This calculation was performed for each patient using the calculator (https://www.cdc.gov/bmi/child-teen-calculator/index.html).
This approach of categorizing children into percentiles based on their BMI values provides a more accurate evaluation method.
https://www.cdc.gov/bmi/child-teen-calculator/bmi-categories.html
As indicated in the results of our study, different outcomes were observed across BMI percentile groups. In line with your suggestion, a detailed explanation of how BMI percentiles are calculated and categorized has been added to the methodology section.
- A lot of discussion bases on BMI-index, which is actually incorrect in children. The Authors should change this parameter.
Answer: Thank you for your comment. The issue you mentioned has been clarified above. It is this more clearly explained in the discussion section as you recommended.
(line 292-310)
Reviewer 2 Report
Comments and Suggestions for Authors
Thank you for permitting me to review this manuscript
In this study the Modified Aldrete Recovery Score (MAS) and an institutional discharge criteria were compared to determine which provides better discharge outcome in case of children having propofol sedation for dental procedures They finded that MAS provided a quicker and equally safe discharge in comparison to the institutional protocol.
Here are my suggestions
it is difficult to say this is a pure sedation as inhalational induction permitting to have an IV access may be considered general anesthesia , therefore some discussion should consider the frontiers between general anesthesia and sedation and the related definition
The title should also include propofol sedation
Please provide a very brief description of MAS scores for the readers not specialized in this area
The results and discussion about the BMI and its effect is a little confusing amd may be re explained
The discussion and conclusion should also consider that the results are valid for propofol sedation and might be different for other types of sedation and other studies might be necessary .
Author Response
- it is difficult to say this is a pure sedation as inhalational induction permitting to have an IV access may be considered general anesthesia , therefore some discussion should consider the frontiers between general anesthesia and sedation and the related definition
Answer: Thank you for your comment. In fact, as seen in the table below, general anesthesia is defined as a condition in which the patient remains unresponsive despite repeated stimuli, spontaneous breathing is impaired often, requiring intubation of the airway, and cardiovascular function is generally impaired. (https://www.asahq.org/standards-and-practice-parameters/statement-on-continuum-of-depth-of-sedation-definition-of-general-anesthesia-and-levels-of-sedation-analgesia) (line 137-141)
In our study, as detailed in the methodology section, inhalation anesthesia is administered at the start of anesthesia to the pediatric patient solely to establish pain-free vascular access. This procedure lasts only a few minutes. During anesthesia induction and maintenance, the patient's spontaneous breathing and cardiovascular function is not allowed to impair. To ensure this, propofol maintenance is administered throughout the procedure while monitoring the patient's sedation status using BIS monitoring. (line 132-136)
In order to avoid confusion a more detailed and clarified description is added to manuscript as above.
- The title should also include propofol sedation :
Answer: Thank you for your valuable contribution. Title is changed and propofol is mentioned in the title as you recommended.: “Assessing Discharge Readiness After Propofol-Mediated Deep Sedation in Pediatric Dental Procedures: Revisiting Discharge Practices with the Modified Aldrete Recovery Score”
- Please provide a very brief description of MAS scores for the readers not specialized in this area.
Answer: Thank you for your contribution. A descriptive table is added to the method section. (Aldrete J. A. (1995). The post-anesthesia recovery score revisited. Journal of clinical anesthesia, 7(1), 89–91. https://doi.org/10.1016/0952-8180(94)00001-k) This was cited as reference 14.
Round 2
Reviewer 1 Report
Comments and Suggestions for Authors
The suggestions had not been truelly adapted, but it could be accepted.